

# Exact correlations in the Lieb–Liniger model and detailed balance out of equilibrium

**J. De Nardis[1], M. Panfil[2*],**

**1** Département de Physique, Ecole Normale Supérieure, PSL Research University, CNRS,
24 rue Lhomond, 75005 Paris, France.
**2** Institute of Theoretical Physics, University of Warsaw,
ul. Pasteura 5, 02-093 Warsaw, Poland.

\* milosz.panfil@fuw.edu.pl

## Abstract

We study the density-density correlation function of the 1D Lieb–Liniger model and obtain an exact expression for the small momentum limit of the static correlator in the thermodynamic limit. We achieve this by summing exactly over the relevant form factors of the density operator in the small momentum limit. The result is valid for any eigenstate, including thermal and non-thermal states. We also show that the small momentum limit of the dynamic structure factors obeys a generalized detailed balance relation valid for any equilibrium state.



# 1  Introduction

One of the most celebrated and experimentally relevant one-dimensional many-body interacting model is the one governed by the Lieb-Liniger Hamiltonian [1]

$$H = -\sum_{i=1}^{N} \partial_{x_i}^2 + 2c \sum_{i>j}^{N} \delta(x_i - x_j) \tag{1}$$

for a system of $N$ bosons with positions $\{x_i\}_{i=1}^N$ on a line of length $L$, interacting point-wise with coupling constant $c$. It is a paradigmatic example of a system of interacting bosons on the continuum and it is experimentally relevant for the physics of elongated clouds of cold atoms with contact interactions [2–7]. As an integrable model it helped to shape the understanding of quantum integrability [8,9] and it represents the non-relativistic limit of many integrable field theories [10–14]. The integrability of the model leads to a complete characterization of the eigenstates of (1) in terms of $N$ quasi-momenta or rapidities $\{\lambda_j\}_{j=1}^N$. On the other hand, and despite this, it is extremely hard to obtain exact predictions for the correlation functions of local operators. Recent years witnessed certain developments in this direction. First the ABACUS method allowed for exact numerical evaluation of the correlation functions in finite systems [15–17]. Second, the low energy limit of the correlation functions was shown to agree with the universal predictions of the Luttinger liquid theory [18,19]. Third, certain one-point correlation functions were derived [10, 11, 20]. Finally, the weakly interacting theory $c \sim 0$ can be studied with a 1D version of the Bogolyubov approximation [21,22], while the strongly interacting limit $c \to \infty$ can be studied perturbatively in $1/c$ [22–24].

Recently new questions have been raised, boosted by the incredible progresses on cold atoms experiments probing the non-equilibrium dynamics of isolated systems. These center around the problem of determining and characterizing the non-equilibrium steady state when the system unitarily evolves from a pure state which is not one of its eigenstates. Some examples are the dynamics induced by an abrupt change of the coupling constant (quantum quench) [25] or by a quick and strong Bragg pulse [26]. It was shown in [27] that for an integrable model as (1) the steady state of a non-equilibrium unitary time evolution (when this is reached) can be represented with a single eigenstate, fixed by the expectation values of the conserved quantities of the Hamiltonian on the initial state. This is called the Generalized-Gibbs-Ensemble (GGE) saddle-point state [28–30] and it determines the expectation values of all the local observables at equilibrium [31–35], and also allows to reconstruct the time evolution towards it [36–39]. These developments open up new challenging problems of computing a correlation function on an arbitrary eigenstate of the system. Progress in this very direction has been recently achieved and one-point functions of local observables are now accessible [11]. Finding a similar closed-form result in the thermodynamic limit for two-point functions is still a challenging task.

In this paper we report a new progress in this direction and obtain an exact expression for the small momentum limit of the static structure factor, the Fourier transform of the two-point density-density correlation function, with the density operator defined as

$$\hat{\rho}(x) = \sum_{j=1}^{N} \delta(x - x_j). \tag{2}$$

We recover well known results in the case of thermal equilibrium gases, which we extend to the case of a generic GGE saddle-point state. We also show that a generalization of the detailed balance to non-thermal equilibrium states is possible in the small momentum regime of density-density correlations. The results are based on our previous work [40] where, for a wide class of eigenstates, we computed the thermodynamic limit of the form factors of the density operator. In particular we have shown that the two-point density-density correlation function can be computed as a sum over all possible particle-hole excitations. In [40] we included only one particle-hole excitations in the computation of the dynamic structure factor $S(k, \omega)$ (the Fourier transform in time and space of the two-point and two-time density correlation functions) and the result was shown to be in good agreement with well established numerical methods. Moreover it was observed that the small momentum limit of the dynamic structure factor is fully given by the one particle-hole contribution. We employ these previous results to compute exactly the small momentum limit of the static density-density correlation function.

In section 2 we summarize the main results. In section 3 we introduce the necessary background on the Lieb-Liniger model and correlation functions in order to derive the small momentum limit of the static structure factor presented in section 4. Some technical aspects of the derivation are described in a greater detail in the appendices B, C and D. In section 5 we show that in the small momentum limit a generalized detailed balance relation holds also for non-thermal states and we use it as a consistency check of the main result. We focus on eigenstates with smooth distribution of the quasi-momenta (finite extensive entropy states) and in appendix A we show that our approach also applies when the distribution function is discontinuous (zero extensive entropy states). Notable examples are the ground state of the Lieb-Liniger model [1] and the split Fermi sea [41].

## 2 Summary of results

We consider the thermodynamic limit of the Lieb-Liniger model: $N \to \infty$ with $n = N/L$ constant, where $L$ is the length of the 1D system and study the density-density correlation on a generic eigenstate $|\vartheta\rangle$ with rapidity density $\rho_p(\lambda)$. This is defined as

$$\bar{S}(x, t) = \langle \vartheta | \hat{\rho}(x, t) \hat{\rho}(0, 0) | \vartheta \rangle - n^2, \tag{3}$$

where $\hat{\rho}(x, t)$ is the density operator[1]. The state $|\vartheta\rangle$ is a state with a finite extensive entropy and it can be either the saddle point (in the thermodynamic limit) of a thermal grand-canonical ensemble or the saddle point state of a more general GGE ensemble[2].

---

[1] We use the hat to distinguish the density operator $\hat{\rho}(x, t)$, a physical operator acting on a Hilbert space, from the density of rapidities $\rho_p(\lambda)$, a function characterizing a state in the Hilbert space

[2] By saddle point we mean the most relevant contribution of a sum over a statistical ensemble in the thermodynamic limit.

The main object of our interest is the static structure factor, this is a Fourier transform of the equal-time correlation function

$$S(k) = \int_{-\infty}^{+\infty} \mathrm{d}x\, e^{ikx} \bar{S}(x, 0), \tag{4}$$

in the small momentum limit, $k \to 0$. We find a closed expression for $S(0)$ in terms of the functions characterizing the averaging state $|\vartheta\rangle$ by summing over the relevant excitations of the gas in the thermodynamic limit. We find

$$S(k) = (2\pi)^2 \int_{-\infty}^{\infty} \mathrm{d}h\, \rho_p(h)\rho_h(h)\rho_t(h) + O(k^2). \tag{5}$$

The integration is performed over the rapidity space and functions $\rho_t(h)$ and $\rho_h(h)$ are uniquelly related to the rapidity density $\rho_p(h)$. The quantity $L\rho_p(h)dh$ specifies the number of particles with rapidity $h$, $L\rho_t(h)dh$ the maximal allowed number of particles with rapidity $h$, and $\rho_h(h) = \rho_t(h) - \rho_p(h)$ is the density of holes. In the non-interacting gas the total density $\rho_t(h)$ is just a constant, here due to interactions, the total available density at rapidity $h$ is coupled to the density of rapidities and it varies with $h$. Finally, for any state $|\vartheta\rangle$ with a finite extensive entropy the corrections for finite $k$ are of order $O(k^2)$.

If we restrict our attention to states representing thermal equilibrium we recover a known results connecting $S(0)$ with the isothermal compressibility $\kappa_T$. The isothermal compressibility describes the change of the density of the gas induced by a change of the chemical potential $\mu$ while keeping the temperature $T$ constant

$$\kappa_T = \left.\frac{dn}{d\mu}\right|_T, \tag{6}$$

where $n$ is the density of the gas and $\mu$ is the chemical potential of the Gibbs ensemble whose partition function is

$$Z_{GE} = \mathrm{tr}\left(e^{-(H-\mu N)/T}\right), \tag{7}$$

with $N$ the particle number operator. The derivative in (6) is taken keeping the temperature fixed. The relation between $S(0)$ and the isothermal compressibility is known by the detailed balance relation [21] and it reads

$$S_T(0) = T\kappa_T. \tag{8}$$

We derive it to be equivalent to our main result (5). The detailed balance relation used here is a relation of dynamic structure factor $S(k, \omega)$ (see eq. (35) for the definition)

$$S_T(k, -\omega) = e^{-\omega/T} S_T(k, \omega), \tag{9}$$

which sets the ratio between the probabilities that the gas absorbs and releases energy $\omega$ due to external perturbation coupling to the density. The generality of expression (5) implies that a relation like (8) holds not only for thermal equilibrium (determined by the Gibbs ensemble) but also for more general equilibrium situations like those determined by the GGE ensemble. In this case the partition function is

$$Z_{GGE} = \mathrm{tr}\left(e^{\mu N + \sum_j \mu_j Q_j}\right), \tag{10}$$

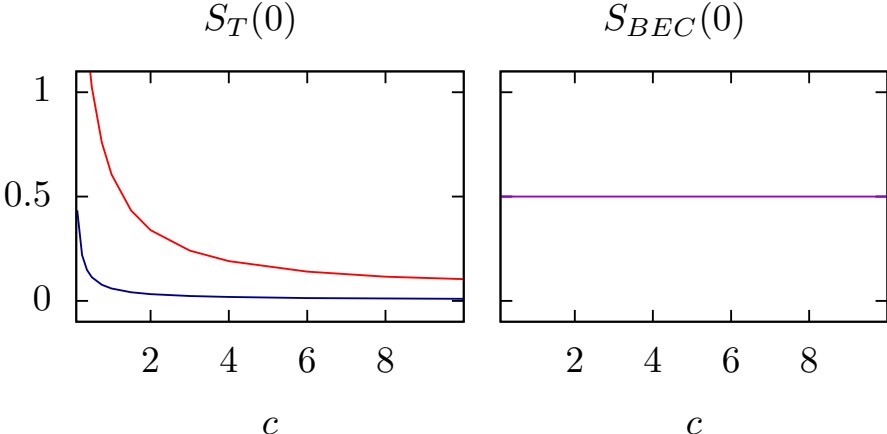

Figure 1: (left) Plot of $S_T(0)$ from equation (8) for a gas of density $n = 1$ at thermal equilibrium as a function of the coupling $c$ and for two different temperatures, $T = 1$ (red line) and $T = 0.1$ (blue line). For high values of the coupling $c$ the particles tend to repel each other and the compressibility $\kappa_T$ tends to be very small, while it diverges in the limit $c \to 0^+$ for any finite temperature $T$. (right) Plot of $S_{\text{BEC}}(0) \equiv S_{\text{GGE}}(0)$ from equation (12) for a gas of density $n = 1$ in the GGE state obtained by quenching the interaction parameter from $c_0 = 0$ (BEC state) to a finite value $c$, as a function of the latter. This state was first determined in [42] and the value $S_{\text{GGE}}(0) = n/2$ (given by equation (12) applied to the relation $n = c\, e^{\mu/2}$ in [42]) for any value of the final coupling $c$ was suggested there by numerical computations (see Fig. 3 in [42]). This result shows that the long time limit of the compressibility after the BEC interaction quench does not depend on the final value of the interaction $c$ but only on the initial state.

where $Q_j$ are local or quasi-local conserved charges and $\mu_j$ the associated chemical potentials (Lagrange multipliers). By analogy with (6) we define generalized isothermal compressibility

$$\kappa_{\text{GGE}} = \frac{dn}{d\mu}\bigg|_{\text{other Lagrange multipliers}}, \tag{11}$$

where the derivative is taken with respect to the chemical potential associated with the particle number and all the other chemical potentials $\{\mu_i\}$ are kept fixed. This turns out to be again related to $S(0)$ through an equivalent relation[3]

$$S_{\text{GGE}}(0) = \kappa_{\text{GGE}}. \tag{12}$$

In figure 1 we present the results for $S(0)$ at the thermal equilibrium and for a specific non-equilibrium gas obtained by the interaction quench [42].

The isothermal compressibility can be used to compute the sound velocity $v_T$ of the system $\kappa_T = 2n/v_T^2$. We can use the same relation to define a generalized sound velocity

$$\kappa_{\text{GGE}} = \frac{2n}{v_{\text{GGE}}^2}, \tag{13}$$

for a GGE state. The GGE sound velocity can be used, for example, as a base for the emergent hydrodynamic description of the non-equilibrium steady states analogously as it is done

---

[3]The factor $T$ in (8) and the lack of it in (12) is due to a rescaling of the chemical potential $\mu$ in the two cases, namely $\mu_{GGE} = \mu_{GE}/T$ with $1/T$ the chemical potential associated to the total energy (which corresponds to the temperature of the gas in a thermal ensemble).

in [43].

We also show that result (12) is consistent with a generalization of the detailed balance (9) to non-thermal state. We argue that for any equilibrium state exists a function $\hat{\mathscr{F}}(k,\omega)$ such that, for small $k$, it generalizes the thermal detailed balance

$$S(k,-\omega) = e^{-\mathscr{F}(k,\omega)}S(k,\omega) + O(k^2), \tag{14}$$

where $\mathscr{F}(k,\omega)$ is defined in (102). This is clearly weaker than its thermal counterpart since it is valid only in the small momentum limit, but it still allows to derive equation (12) from standard hydrodynamic arguments as it provides a generalization of the compressibility sum rule [21]

$$\lim_{k\to 0}\int_{-\infty}^{\infty}\frac{\mathrm{d}\omega}{2\pi}\frac{S_{\mathrm{GGE}}(k,\omega)}{\hat{\mathscr{F}}(k,\omega)} = \frac{\kappa_{\mathrm{GGE}}}{2}. \tag{15}$$

We claim that similar relations hold for any integrable model with particle-hole excitations and where the one particle-hole excitations are the dominant ones in the small $k$ regime.

## 3 The Lieb-Liniger model and correlation functions

The Hamiltonian of the 1D Bose gas (Lieb-Liniger model) is

$$H = -\sum_{i=1}^{N}\partial_{x_i}^2 + 2c\sum_{i>j}^{N}\delta(x_i-x_j). \tag{16}$$

Here $x_i$ denotes the position of the $i$th particle and we choose to work in unites where $\hbar^2/2m = 1$. We consider the gas to be confined to a finite length $L$ with periodic boundary conditions. The effective interaction is characterized by a parameter $\gamma = c/n$ where $n = N/L$ is the 1D density. The eigenstates are fully characterized by rapidities $\boldsymbol{\lambda} = \{\lambda_j\}_{j=1}^N$, which due to the periodic boundary conditions, solve the Bethe equations

$$\lambda_j + \sum_{k=1}^{N}\theta(\lambda_j-\lambda_k) = \frac{2\pi}{L}I_j, \qquad j = 1,\dots,N, \tag{17}$$

where $\theta(\lambda) = 2\arctan(\lambda/c)$ is the two-particle phase shift and $\{I_j\}_{j=1}^N$ are quantum numbers which uniquely label the eigenstates. The quantum numbers are even (for N odd) or half-odd (for N odd) integers and obey the Pauli principle. The energy and momentum are

$$E[\boldsymbol{\lambda}] = \sum_{j=1}^{N}\epsilon^{(2)}(\lambda_j), \qquad P[\boldsymbol{\lambda}] = \sum_{j=1}^{N}\epsilon^{(1)}(\lambda_j). \tag{18}$$

where

$$\epsilon^{(j)}(\lambda) = \lambda^j, \tag{19}$$

is the bare eigenvalue of the ultra-local conserved charge $Q_j$[4] of the model (1), where $Q_1 = P$ momentum operator and $Q_2 = H$ the Hamiltonian. The set of charges with eigenvalues $\lambda^j$ (ultra-local charges) can be replaced by other sets as the one introduced in [44] (semi-local charges) which are more relevant for certain non-equilibrium situations. From now on we will

---

[4] By conserved charge we refer to an operator $Q$ that commutes with the Hamiltonian (1). An ultra-local conserved charge is an operator whose density acts only on a single point $x \in \mathbb{R}$, while a semi local charge acts on a finite interval of $\mathbb{R}$.

denote with $\{\epsilon^{(j)}(\lambda)\}_j$ a generic set of eigenvalues of conserved charges of the Hamiltonian (1).

In the thermodynamic limit, $L \to \infty$ with $n = N/L$ fixed [45], the Bethe states are characterized by a filling function $\vartheta(\lambda)$ which is a ratio between the number of rapidities in the interval $(\lambda, \lambda + d\lambda)$ and the maximal number of them allowed in such interval. The filling function naturally obeys

$$0 \leq \vartheta(\lambda) \leq 1. \tag{20}$$

and it can have finite or infinite support. Our main result is valid for any differentiable filling function $\vartheta(\lambda)$ with support on the whole real axis[5]. At equilibrium the filling function $\vartheta(\lambda)$ follows from constraints imposed on the system. In particular it takes an universal form

$$\vartheta(\lambda) = \frac{1}{1 + e^{\epsilon(\lambda)}}, \tag{21}$$

and $\epsilon(\lambda)$ solves an integral equation, known as generalized thermodynamic Bethe ansatz equation [27, 45–47]

$$\epsilon(\lambda) = \sum_j \mu_j \epsilon^{(j)}(\lambda) - \mu - \frac{1}{2\pi} \int_{-\infty}^{\infty} d\lambda' K(\lambda - \lambda') \log(1 + e^{-\epsilon(\lambda')}) \tag{22}$$

where $\mu$ is the chemical potential associated to the total density $n$ of the gas and $\mu_j$ is a chemical potential associated to conserved charge $Q_j$. The kernel $K(\lambda)$ is given by

$$K(\lambda) = \frac{2c}{\lambda^2 + c^2}. \tag{23}$$

The case of a pure thermal gas is given by $\mu_2 = \frac{1}{T}$ with $\epsilon^{(2)}(\lambda) = \lambda^2$ and with all the other $\mu_j$ equal to zero. An example of a non-thermal equilibrium gas is a state reached after the quench from the BEC state, the ground state of the non-interacting gas $c = 0$. From the results in [42] the saddle point distribution $\vartheta_{\mathrm{BEC}}(\lambda)$ characterizing the gas in the long time after the quench is known[6]. The $\vartheta_{\mathrm{BEC}}(\lambda)$ can be expressed as

$$\vartheta_{\mathrm{BEC}} = \frac{a(\lambda/c)}{a(\lambda/c) + 1}, \qquad a(x) = \frac{2\pi\tau}{x \sinh(2\pi x)} I_{1-2ix}(4\sqrt{\tau}) I_{1+2ix}(4\sqrt{\tau}). \tag{24}$$

Once the filling function $\vartheta$ is specified all the other functions characterizing the states, like the distribution of rapidities $\rho_p(\lambda)$, are fixed. The interacting nature of the gas implies that the maximal density of the particles (that we denote $\rho_t(\lambda)$) is not constant and is connected with the filing function through an integral equation

$$\rho_t(\lambda) = \frac{1}{2\pi} + \int_{-\infty}^{\infty} d\lambda' \frac{\vartheta(\lambda')}{2\pi} K(\lambda - \lambda') \rho_t(\lambda'), \tag{25}$$

which is obtained by taking the thermodynamic limit of (17). The density of rapidities is given by

$$\rho_p(\lambda) = \vartheta^{-1}(\lambda) \rho_t(\lambda), \tag{26}$$

---

[5]However we show in the appendix A that this constraint can be relaxed and our expression for $S(0)$, together with its first order correction in $k$, is valid also for discontinuous functions as the ground state of the system.

[6]While in [42] it was determined via the quench action method, this is equivalent to fix the distribution via a complete set of quasi-local conserved charges.

and is normalized by the total density

$$\int_{-\infty}^{\infty} d\lambda\, \rho_p(\lambda) = n. \tag{27}$$

The density of rapidities determines all the macroscopic variables of the state, like the energy and the momentum density

$$\frac{E[\vartheta]}{L} = \int_{-\infty}^{\infty} d\lambda\, \rho_p(\lambda)\lambda^2, \qquad \frac{P[\vartheta]}{L} = \int_{-\infty}^{\infty} d\lambda\, \rho_p(\lambda)\lambda. \tag{28}$$

With $|\vartheta\rangle$ we denote a macroscopic state with a filling function $\vartheta(\lambda)$ and associated through (26) the density function $\rho_p(\lambda)$. Note that there are many microstates (specified by different sets of rapidities $|\boldsymbol{\lambda}\rangle$) that correspond to the same thermodynamic state. Their extensive number is given by the Yang-Yang entropy [45]

$$S_{YY}[\vartheta] = L \int_{-\infty}^{+\infty} d\lambda \left( \rho_t(\lambda)\log(\rho_t(\lambda)) - \rho_p(\lambda)\log(\rho_p(\lambda)) - \rho_h(\lambda)\log(\rho_h(\lambda)) \right). \tag{29}$$

Therefore when the state $|\vartheta\rangle$ has a finite extensive entropy we have $S_{YY}[\vartheta] > 0$.

In studies of correlation functions it is important to understand the structure of the excitations around a given macroscopic state $|\vartheta\rangle$. The excitations are particle or holes in the density function. The excitations are of positive (particles) or negative (holes) energy. The density operator conserves number of the particles and therefore we consider only pairs of particle-hole excitations. We can write a density of an excited macroscopic state (around $|\vartheta\rangle$) as

$$\rho_\lambda(\lambda) = \rho_p(\lambda) + \frac{1}{L}\sum_{j=1}^{m}\delta(\lambda - p_j) - \frac{1}{L}\sum_{j=1}^{m}\delta(\lambda - h_j) + O(L^{-2}), \tag{30}$$

where $m$ stands for the number of particle-hole excitations. We denote such an excited state by $|\vartheta; \mathbf{p}, \mathbf{h}\rangle$ using a notation $\mathbf{p} \equiv \{p\}$. The momentum and energy of an excited state relative to the averaging state are

$$k(\vartheta, \mathbf{p}, \mathbf{h}) = \sum_{j=1}^{m} p_j - \sum_{j=1}^{m} h_j - \int_{-\infty}^{+\infty} d\lambda\, \vartheta(\lambda) F(\lambda|\mathbf{p}, \mathbf{h}), \tag{31}$$

$$\omega(\vartheta, \mathbf{p}, \mathbf{h}) = \sum_{j=1}^{m} p_j^2 - \sum_{j=1}^{m} h_j^2 - 2\int_{-\infty}^{+\infty} d\lambda\, \vartheta(\lambda)\lambda F(\lambda|\mathbf{p}, \mathbf{h}), \tag{32}$$

where the back-flow function satisfies

$$F(\lambda\,|\,\mathbf{p}, \mathbf{h}) = \sum_{j=1}^{m} F(\lambda\,|\,p_j, h_j), \tag{33}$$

$$2\pi F(\lambda\,|\,p, h) = \theta(\lambda - p) - \theta(\lambda - h) + \int_{-\infty}^{\infty} d\lambda'\, K(\lambda - \lambda')\vartheta(\lambda') F(\lambda'\,|\,p, h), \tag{34}$$

and it describes a deformation of the rapidity distribution of the averaging state due to a particle-hole excitation. Namely, after a single particle hole excitation $h \to p$, the rapidities $\lambda_j$ get shifted as $\lambda_j \to \lambda_j - F(\lambda_j|p, h)/L$.

These type of excitations are known to be the only one relevant in the thermodynamic limit. They are responsible for the local correlations of the gas in the thermodynamic limit [18, 48], transport in non-equilibrium systems [43, 49], post-quench entanglement evolution [50] and approach to equilibrium of expectation values during a non-equilibrium time evolution [27, 37, 51].

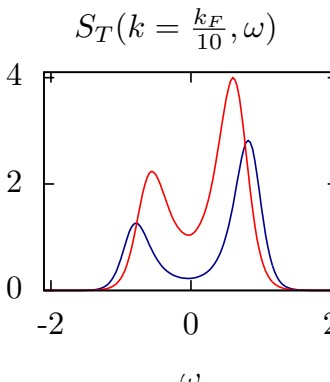
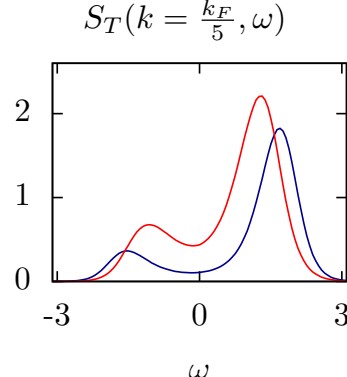

Figure 2: Plot of $S_T(k, \omega)$ at fixed $k$ obtained from a numerical evaluation of the $m = 1$ contribution (one particle-hole excitations) of equation (38) with $k = k_F/10$ (left) and $k = k_F/5$ (right) ($k_F = \pi n$) for a thermal gas with $T = 1$, $n = 1$ and for two different values of the coupling $c = 1$ (red line) and $c = 2$ (blue line).

### 3.1 Dynamic structure factor and one particle-hole excitations

We consider Fourier transform of $\bar{S}(x, t)$ (3), the dynamic structure factor

$$S(k, \omega) = \int_0^L \mathrm{d}x \int_{-\infty}^{\infty} \mathrm{d}t\, e^{i(kx - \omega t)} \bar{S}(x, t) \tag{35}$$

which satisfies the f-sum rule [21]

$$\frac{1}{nk^2} \int_{-\infty}^{\infty} \frac{\mathrm{d}\omega}{2\pi} \omega S(k, \omega) = 1, \tag{36}$$

that holds for an arbitrary state $|\vartheta\rangle$ and is a manifestation of the conservation of the particles number. The main object of our interest is the static correlator (4) which in terms of the dynamic structure factor is

$$S(k) = \int_{-\infty}^{\infty} \frac{\mathrm{d}\omega}{2\pi} S(k, \omega). \tag{37}$$

The dynamic structure factor in the thermodynamic limit and in the spectral representation can be written schematically as [40]

$$S(k, \omega) = (2\pi)^2 \sum_{m=1}^{\infty} \sum_{(\mathbf{p},\mathbf{h}) \in \mathscr{H}_\vartheta^{(m)}} |\langle \vartheta | \hat{\rho}(0) | \vartheta, \mathbf{h} \to \mathbf{p} \rangle|^2 \delta(k - k(\vartheta, \mathbf{p}, \mathbf{h})) \delta(\omega - \omega(\vartheta, \mathbf{p}, \mathbf{h})). \tag{38}$$

Here $\mathscr{H}_\vartheta^{(m)}$ is a Hilbert space of $m$-particle-hole excitations around the averaging state $|\vartheta\rangle$. The sum is organized in number of particle-hole excitations on the state $|\vartheta\rangle$. Each sum is an $m^2$-dimensional integral in the space of rapidities and $\langle \vartheta | \hat{\rho}(0) | \vartheta, \mathbf{h} \to \mathbf{p} \rangle$ are the form factor of the density operator between the state $|\vartheta\rangle$ and the state $|\vartheta, \mathbf{h} \to \mathbf{p}\rangle$ with a number $m$ of particle-hole excitations. They are reported in (42) and one should note that they are non-trivial functions of $\vartheta(\lambda)$ and of the positions of the particle-holes $\mathbf{p}, \mathbf{h}$.

In [40] was found that already the first term of the sum $m = 1$ provides a good approximation. In fig. 2 we show the one particle-hole contribution to $S(k, \omega)$ and in fig. 3 we show the one particle-hole contribution to the f-sum rule and the static correlator for small values of $k$. The results suggests that at small momenta the correlation function is saturated by the single

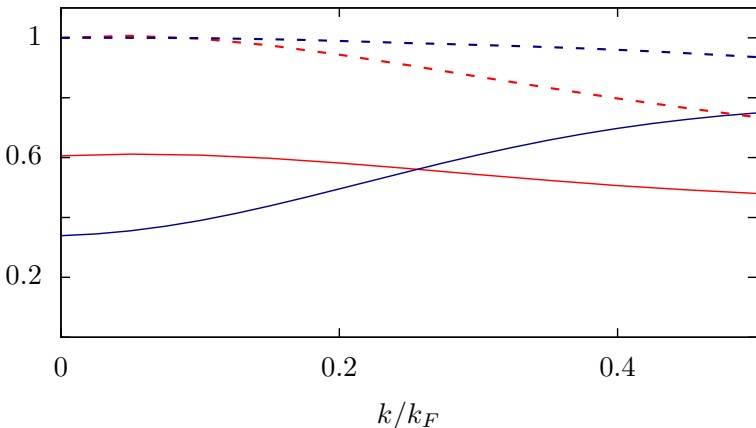

Figure 3: Plot of the static density factor $S_T(k)$ (solid lines) at thermal equilibrium (with temperature $T = 1$ and density $n = 1$) at different values of the coupling $c$, obtained from a numerical evaluation of the one particle-hole contribution (corresponding to the contribution $m = 1$ in (38)) and the related f-sum rules (39) (dashed lines). At $c = 1$ (red lines) the f-sum rule decays from the expected value 1 quite fast as $k$ increases, signaling that one particle-hole do not saturate the sum (38) for higher momenta, while at $c = 2$ (blue lines) the f-sum rule decays much slower as $k$ increases. In both cases the limit $k \to 0$ is exactly given by the one particle-hole contribution as both sum rules converge in this limit to their exact value 1 and the static density factors converge to their exact thermal values given by (8). We conclude that the small momentum part of $S(k, \omega)$ in (38) is then only given by the one particle-hole contribution $m = 1$ for any value of the interaction $c \geq 0$ up to corrections $O((k/c)^s)$ with $s > 1$.

particle-hole excitation. We check this using the f-sum rule (36). Indeed for a thermal state we can use the detailed balance and rewrite the f-sum rule (36) as a sum of positive terms

$$\int_0^\infty \frac{d\omega}{2\pi} \omega S_T(k, \omega)(1 - e^{-\omega/T}) = nk^2. \tag{39}$$

Therefore, since each contribution in the sum (38) is positive, it provides a good measure of the convergence of the sum in (38). We can then write an exact expression of $S(k, \omega)$ in the small momentum limit involving only a sum over one particle-excitations

$$S(k \sim 0, \omega) =$$
$$(2\pi)^2 \int_{-\infty}^{+\infty} dp \rho_h(p) \int_{-\infty}^{+\infty} dh \rho_p(h) |\langle \vartheta | \hat{\rho}(0) | \vartheta, h \to p \rangle|^2 \delta(k - k(p, h)) \delta(\omega - \omega(p, h)) \tag{40}$$

where corrections are expected to be of $O((k/c)^2)$. This leads to the following expression for the static correlation function at zero momentum

$$S(0) = 2\pi \lim_{k \to 0} \int_{-\infty}^{+\infty} dp \rho_h(p) \int_{-\infty}^{+\infty} dh \rho_p(h) |\langle \vartheta | \hat{\rho}(0) | \vartheta, h \to p \rangle|^2 \delta(k - k(p, h)). \tag{41}$$

This expression is the starting point of the computation.

## 3.2  Thermodynamic limit of the form-factor

We recall here the formula for the form factor of the density operator $\hat{\rho}$ in the thermodynamic limit. This formula was derived in [40]

$$
\begin{aligned}
|\langle\vartheta|\hat{\rho}(0)|\vartheta,\{h_j\to p_j\}_{j=1}^m\rangle| &= \frac{c}{2}\left[\prod_{k=1}^m \frac{F(h_k)}{(\rho_t(p_k)\rho_t(h_k))^{1/2}}\frac{\pi\tilde{F}(p_k)}{\sin\pi\tilde{F}(p_k)}\frac{\sin\pi\tilde{F}(h_k)}{\pi\tilde{F}(h_k)}\right] \\
&\times \prod_{i,j=1}^m\left[\frac{(p_i-h_j+ic)^2}{(h_{i,j}+ic)(p_{i,j}+ic)}\right]^{1/2}\frac{\prod_{i<j=1}^m h_{i,j}p_{i,j}}{\prod_{i,j=1}^m(p_i-h_j)}\det_{i,j=1}^m\left(\delta_{ij}+W(h_i,h_j)\right) \\
&\times \exp\left(-\frac{1}{4}\int_{-\infty}^{+\infty}d\lambda d\lambda\left(\frac{\tilde{F}(\lambda)-\tilde{F}(\lambda')}{\lambda-\lambda'}\right)^2-\frac{1}{2}\int_{-\infty}^{+\infty}d\lambda d\lambda'\left(\frac{\tilde{F}(\lambda)\tilde{F}(\lambda')}{(\lambda-\lambda'+ic)^2}\right)\right) \\
&\times \exp\left(\sum_{k=1}^m \mathrm{PV}\int_{-\infty}^{+\infty}d\lambda\frac{\tilde{F}(\lambda)(h_k-p_k)}{(\lambda-h_k)(\lambda-p_k)}+\int_{-\infty}^{+\infty}d\lambda\frac{\tilde{F}(\lambda)(p_k-h_k)}{(\lambda-h_k+ic)(\lambda-p_k+ic)}\right) \\
&\times \frac{\mathrm{Det}(1+\hat{A})}{\mathrm{Det}(1-\frac{\vartheta}{2\pi}K)}\exp\left(\sum_{j=1}^m\delta S[\vartheta;p_j,h_j]\right).
\end{aligned} \tag{42}
$$

The $\vartheta$ is the filling function characterizing an eigenstate, $F(\lambda)=F(\lambda|\mathbf{p},\mathbf{h})$ is the back-flow function (34) and we also defined a rescaled back-flow $\tilde{F}(\lambda)=\vartheta(\lambda)F(\lambda)$ to shorten the formula. The two determinants of the last line are Fredholm determinants. The kernel $A$ is

$$
A(\lambda,\lambda')=\tilde{a}(\lambda)\left(K(\lambda-\lambda')-\frac{2}{c}\right) \tag{43}
$$

where

$$
\begin{aligned}
\tilde{a}(\lambda)=&-\frac{\sin[\pi\vartheta(\lambda)F(\lambda)]}{2\pi\sin[\pi F(\lambda)]}\frac{\prod_{k=1}^m p_k-\lambda}{\prod_{\substack{k=1 \\ h_k\neq\lambda}}^m h_k-\lambda}\prod_{k=1}^n\left(\frac{K(p_k-\lambda)}{K(h_k-\lambda)}\right)^{1/2} \\
&\times \exp\left[-\frac{c}{2}\mathrm{PV}\int d\lambda'\frac{\vartheta(\lambda')F(\lambda')K(\lambda'-\lambda)}{\lambda'-\lambda}\right].
\end{aligned} \tag{44}
$$

Function $W(h,\lambda)$ is a solution to the following linear integral equation (to be solved on the second variable $\lambda$)

$$
W(h,\lambda)+\int_{-\infty}^{\infty}d\alpha\,W(h,\alpha)\tilde{a}(\alpha)\left(K(\alpha-\lambda)-\frac{2}{c}\right)=b(h)\left(K(h-\lambda)-\frac{2}{c}\right), \tag{45}
$$

with the vector $b(h)$ given by

$$
b(h)=-\frac{\tilde{a}(h)}{\vartheta(h)F(h)}. \tag{46}
$$

Function $\delta S[\vartheta;p,h]$ is a differential entropy defined as

$$
\delta S[\vartheta;p,h]=\int_{-\infty}^{\infty}d\lambda\,s[\vartheta;\lambda]\frac{\partial}{\partial\lambda}\left(\frac{F(\lambda|p,h)}{\rho_t(\lambda)}\right), \tag{47}
$$

where $s[\vartheta;\lambda]$ is the entropy density expressible through the density functions (see eq. (29))

$$
s[\vartheta;\lambda]=\rho_t(\lambda)\log\rho_t(\lambda)-\rho_p(\lambda)\log\rho_p(\lambda)-\rho_h(\lambda)\log\rho_h(\lambda). \tag{48}
$$

# 4 Small momentum limit of the static structure factor

In this section we derive the small momentum limit of a single particle-hole form factor of the density operator (42) and use it to compute the small momentum limit of the static structure factor $S(k)$ from eq. (41). The crucial steps in the derivation involve analysis of certain linear integral equations. Therefore we start this section by recalling the notion of the resolvent.

We will be dealing with linear integral equations of the following form

$$f(\lambda) - \int_{-\infty}^{\infty} d\lambda' \frac{\vartheta(\lambda')}{2\pi} K(\lambda - \lambda') f(\lambda') = g(\lambda), \tag{49}$$

with different driving functions $g(\lambda)$. Analysis of families of such equations is greatly simplified if we introduce a resolvent $L(\lambda, \lambda')$ of the kernel $K(\lambda, \lambda')$. With the help of the resolvent the solution to (49) is

$$f(\lambda) = g(\lambda) + \int_{-\infty}^{+\infty} d\lambda' L(\lambda, \lambda') g(\lambda'). \tag{50}$$

In appendix B we give all the necessary information on the resolvent. Among other things we obtain the following expressions for the total density $\rho_t(\lambda)$ and the back-flow $F(\lambda|p, h)$

$$\rho_t(\lambda) = \frac{1}{2\pi} \left( 1 + \int_{-\infty}^{+\infty} d\alpha L(\lambda, \alpha) \right), \tag{51}$$

$$F(\lambda|p, h) = -\frac{1}{\vartheta(\lambda)} \int_h^p d\alpha L(\alpha, \lambda). \tag{52}$$

We also note that the resolvent obeys the following symmetry relation under exchanging its arguments

$$\vartheta(\lambda') L(\lambda', \lambda) = \vartheta(\lambda) L(\lambda, \lambda'). \tag{53}$$

## 4.1 One particle-hole kinematics at low momentum

We start by analysing the structure of single small momentum particle-hole excitations. The momentum and the energy, according to eqs. (31) and (32), are

$$k = p - h - \int_{-\infty}^{+\infty} d\lambda \vartheta(\lambda) F(\lambda|p, h), \tag{54}$$

$$\omega = p^2 - h^2 - 2 \int_{-\infty}^{+\infty} d\lambda \vartheta(\lambda) \lambda F(\lambda|p, h). \tag{55}$$

We take the difference of the rapidities $p - h$ to be small which allows us to approximate the integral in (52) and obtain

$$F(\lambda|p, h) = -(p - h)\vartheta^{-1}(\lambda) L(h, \lambda). \tag{56}$$

This leads to the following expression for the momentum

$$k = (p - h)\left( 1 + \int_{-\infty}^{+\infty} d\lambda L(h, \lambda) \right) = 2\pi(p - h)\rho_t(h), \tag{57}$$

where we used a relation (51) between the resolvent and the total density. For the energy we find a linear dispersion relation

$$\omega = v(h)k, \tag{58}$$

with a sound velocity that depends on a position of the excitation

$$v(h) = \frac{h + \int_{-\infty}^{+\infty} d\lambda \lambda L(h, \lambda)}{\pi \rho_t(h)} \equiv \frac{d\omega(k)}{dk}. \tag{59}$$

The velocity can be both positive and negative. The excitations with large values of $h$, with diverging velocity $v(h) \sim h$, are suppressed by the vanishing density particles $\rho_p(h)$.

## 4.2 Small momentum limit of the form factors

We start now computations of the small momentum limit of the density form factor (42). We consider only single particle-hole excitations, $m = 1$ (see Fig. 3). We proceed in two steps. First let us look at the terms that have a simple limit. In the second step we compute the small momentum limit of more involved terms like Fredholm determinants and function $W(h, h)$. An example of a term that has a simple limit is the following product

$$\prod_{i,j}^{n} \left[ \frac{(p_i - h_j + ic)^2}{(h_{i,j} + ic)(p_{i,j} + ic)} \right]^{1/2} = \frac{(p - h + ic)}{ic} = 1 + \mathscr{O}(k). \tag{60}$$

To simplify other terms we observe that the back-flow function is of order $k$. There is a number of simplifications. The ratios of the back-flow functions are

$$\frac{\pi \tilde{F}(p_k)}{\sin \pi \tilde{F}(p_k)} = 1 + \mathscr{O}(k^2). \tag{61}$$

and some integrals over the back-flow also have simple limits

$$\exp\left( -\frac{1}{4} \int_{-\infty}^{+\infty} d\lambda d d\lambda' \left( \frac{\tilde{F}(\lambda) - \tilde{F}(\lambda')}{\lambda - \lambda'} \right)^2 - \frac{1}{2} \int_{-\infty}^{+\infty} d\lambda d\lambda' \left( \frac{\tilde{F}(\lambda)\tilde{F}(\lambda')}{(\lambda - \lambda' + ic)^2} \right) \right) = 1 + \mathscr{O}(k^2) \tag{62}$$

$$\exp\left( \sum_{k=1}^{n} \mathrm{PV} \int_{-\infty}^{+\infty} d\lambda \frac{\tilde{F}(\lambda)(h_k - p_k)}{(\lambda - h_k)(\lambda - p_k)} + \int_{-\infty}^{+\infty} d\lambda \frac{\tilde{F}(\lambda)(p_k - h_k)}{(\lambda - h_k + ic)(\lambda - p_k + ic)} \right) = 1 + \mathscr{O}(k). \tag{63}$$

Finally, the differential entropy $\delta S[\vartheta, p, h]$ (47) is also proportional to the momentum in the small momentum limit and

$$\exp(\delta S[\vartheta, p, h]) = 1 + \mathscr{O}(k). \tag{64}$$

Combining together all these simplifications we obtain a much more compact expression for the form factor

$$|\langle \vartheta | \hat{\rho} | \vartheta, h \to p \rangle| = \frac{c}{2} \frac{1}{(\rho_t(p)\rho_t(h))^{1/2}} \frac{F(h)}{(p - h)} (1 + W(h, p)) \frac{\mathrm{Det}(1 + \hat{A})}{\mathrm{Det}(1 - \frac{\vartheta}{2\pi}K)} \times (1 + \mathscr{O}(k)). \tag{65}$$

The prefactor simplifies

$$\frac{1}{(\rho_t(p)\rho_t(h))^{1/2}} \frac{F(h)}{(p - h)} = \frac{L(h, h)}{\rho_p(h)} + \mathscr{O}(k), \tag{66}$$

where we used that the density function is smooth so $\rho_t(p) = \rho_t(h) + \mathscr{O}(k)$. This gives us

$$|\langle \vartheta | \hat{\rho} | \vartheta, h \to p \rangle| = \frac{c}{2} \frac{L(h, h)}{\rho_p(h)} (1 + W(h, h)) \frac{\mathrm{Det}(1 + \hat{A})}{\mathrm{Det}(1 - \frac{\vartheta}{2\pi}K)} \times (1 + \mathscr{O}(k)). \tag{67}$$

To proceed further we need to study the Fredholm determinants and the integral equation for function $W(h, \lambda)$. They both involve kernel $A(\lambda, \lambda')$ and we start with it. The small momentum limit of the prefactor $\tilde{a}(\lambda)$ is straightforward. The only complication is a need to distinguish $\lambda \neq h$ case from the $\lambda = h$. The result is

$$\tilde{a}(\lambda) = -\frac{\vartheta(\lambda)}{2\pi}, \quad \lambda \neq h, \tag{68}$$

$$\tilde{a}(h) = -\frac{\vartheta(h)}{2\pi}(p - h). \tag{69}$$

Integration is not influenced by a single discontinuity of the integrand and therefore for the kernel $A(\lambda, \lambda')$ we can simply take

$$A(\lambda, \lambda') = -\frac{\vartheta(\lambda)}{2\pi}\left(K(\lambda, \lambda') - \frac{2}{c}\right), \tag{70}$$

for all $\lambda \in \mathbb{R}$. This is a crucial simplification because we are now able to use the same resolvent $L(\lambda, \lambda')$ in analysis of the Fredholm determinants and $W(h, \lambda)$ as we used in solving formally for the $\rho_t(\lambda)$ and the back-flow. On the other hand, the discontinuity affects the driving term $b(h)$ of the integral equation for $W(h, \lambda)$ where we need to take $\tilde{a}(\lambda)$ exactly at $\lambda = h$. For the Fredholm determinant we get

$$\text{Det}\left(1 + \hat{A}\right) = \text{Det}\left(1 - \frac{\vartheta}{2\pi}\left(K - \frac{2}{c}\right)\right). \tag{71}$$

This Fredholm determinant is a simple rescaling of the Gaudin determinant. In the appendix C we show that

$$\text{Det}\left(1 - \frac{\vartheta}{2\pi}\left(K - \frac{2}{c}\right)\right) = \left(1 + \frac{2n}{c}\right)\text{Det}\left(1 - \frac{\vartheta}{2\pi}K\right). \tag{72}$$

To write an equation for $W(h, \lambda)$ we still need to take the small momentum limit of the driving term. The function $b(h)$, according to (69) and (56), becomes

$$b(h) = -\frac{\tilde{a}(h)}{\vartheta(h)F(h)} = \frac{p - h}{2\pi F(h)} = -\frac{\vartheta(h)}{2\pi L(h, h)}. \tag{73}$$

The equation for $W(h, \lambda)$ is now

$$W(h, \lambda) - \int_{-\infty}^{\infty} d\alpha \frac{\vartheta(\alpha)}{2\pi} W(h, \alpha)\left(K(\alpha - \lambda) - \frac{2}{c}\right) = b(h)\left(K(h - \lambda) - \frac{2}{c}\right). \tag{74}$$

This equation can be solved in terms of the total density and the back-flow function. The result is quite complicated but it turns out that a number of simplifications occurs such that the final formula for the form-factor is very simple. For the function $W(h, h)$ we find (for the derivation we refer to the appendix D)

$$W(h, h) = -1 + 2\pi \rho_t(h)\rho_p(h)\frac{2}{c}\left(1 + \frac{2n}{c}\right)^{-1} L^{-1}(h, h). \tag{75}$$

Combining this result and the expression for the Fredholm determinant with the intermediate expression (67) for the form factor we get the final answer

$$|\langle \vartheta|\hat{\rho}|\vartheta, h \to p\rangle| = 2\pi \rho_t(h) \times (1 + \mathcal{O}(k)). \tag{76}$$

The case of two particle-hole excitations $m = 2$ is obviously more complicated. However it can be argued that higher order particle-hole excitations enter with higher powers of $k$. If we assume that the main contribution to $S(k, \omega)$ comes from a region where $\omega \sim k$ then the product of shift functions computed at the position of the holes $\prod_{i=1}^{m} F(h_i)$ in (42) is always proportional to $k^m$ in the small $k$ limit. That is why in order to compute the $O(k)$ of $S(k, \omega)$ from (38) we can neglect the contribution from higher order particle-hole excitations with $m > 1$ (as shown in Fig. 3).

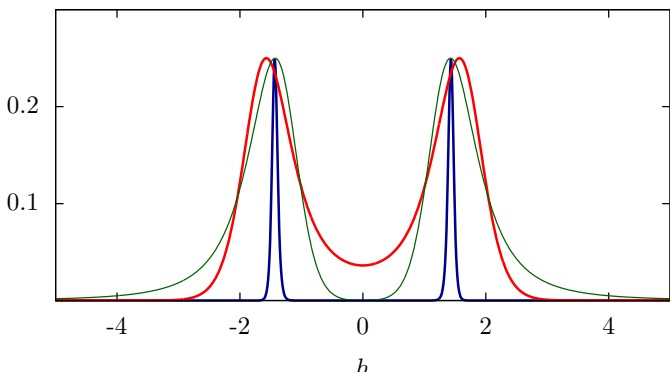

Figure 4: Plot of $(2\pi)^2\rho_p(h)\rho_h(h)\rho_t(h)$ as in formula (79) which corresponds to the weight of the zero momentum one particle-hole excitations. The plot is done for coupling constant $c = 1$ and $n = 1$ and for a state at temperature $T = 0.1$ (blue line), $T = 1$ (red line) and the GGE state given by the BEC quench (green line). At small $T$ (blue line) only excitations performed at the edges of the Fermi sea (the so called Fermi points) have a finite weight. As the state acquires temperature (red line) or in general entropy, as in the BEC state (green line), zero momentum excitations can be created in much wider window of $h$. Therefore states with larger entropy have larger values of $S(0)$, as shown in Fig. 1.

### 4.3 Static structure factor and compressibility of the gas

We recall the eq. (41) for the static correlator in the small momentum limit

$$S(k \sim 0) = 2\pi \int_{-\infty}^{+\infty} dp\,\rho_h(p) \int_{-\infty}^{+\infty} dh\,\rho_p(h)|\langle\vartheta|\hat{\rho}(0)|\vartheta, h \to p\rangle|^2 \delta(k - k(p,h)). \quad (77)$$

The delta function can be resolved using (57) relating momentum to the particle-hole position and valid for small momenta

$$\delta(k - k(p,h)) = \delta(k - 2\pi(p - h)\rho_t(h)) = \frac{1}{2\pi\rho_t(h)}\delta\left((p - h) - \frac{k}{2\pi\rho_t(h)}\right). \quad (78)$$

Using eq. (76) for the form factor we find that the static correlator in the limit $k \to 0$ is given by

$$S(k) = (2\pi)^2 \int_{-\infty}^{+\infty} dh\,\rho_h(h)\,\rho_p(h)\rho_t(h) + O(k^2). \quad (79)$$

This expression is valid for an arbitrary averaging state with a differentiable filling function $\vartheta(\lambda)$. The restriction to the differentiable filling functions comes from the expression for the form factor which was derived under this assumption. If the filling function is not differentiable than one has to consider a subtle structure of the form factor in the vicinity of the discontinuities which qualitatively changes their thermodynamic limit [18,48]. However it turns out that despite that the form factors depend strongly on differentiability of the filling function, the resulting correlation function does not. In the case of the ground state, the product of densities $\rho_h(h)\rho_p(h)$ is a finite number only at the two Fermi edges, therefore the integral (79) gives $\lim_{T\to0} S_T(0) = 0$ as expected, see Fig. 4. This result is actually valid for any distribution with particle and hole densities with separate supports (which are states with vanishing extensive entropy). For more details on the correlations of the ground state $T \to 0$, see section A.

If the gas is in an equilibrium described by a Gibbs ensemble or its generalized version then $S(0)$ can be connected with the isothermal compressibility as in (8). For a generalized Gibbs ensemble (GGE) this is a new result. We derive this correspondence for both cases at once.

The derivative of the generalized dressed energy of equation (22) with respect to the chemical potential $\mu$ (and keeping all the other chemical potentials fixed) is given by

$$\frac{\partial \epsilon(\lambda)}{\partial \mu}\bigg|_{\text{other Lagrange multipliers}} = -2\pi \rho_t(\lambda), \tag{80}$$

which leads to the following relation

$$\frac{\partial \vartheta(\lambda)}{\partial \mu}\bigg|_{\text{other Lagrange multipliers}} = 2\pi \vartheta(\lambda)\rho_h(\lambda). \tag{81}$$

This equality allows to derive the following integral equation for the derivative of $\rho_p(\lambda)$ with respect to the chemical potential

$$\vartheta^{-1}(\lambda)\frac{\partial \rho_p(\lambda)}{\partial \mu} = 2\pi \rho_p(\lambda)\rho_h(\lambda) + \frac{1}{2\pi}\int_{-\infty}^{\infty} d\alpha K(\lambda, \alpha)\frac{\partial \rho_p(\alpha)}{\partial \mu}. \tag{82}$$

This equation follows from rewriting eq. (25) connecting $\rho_t(\lambda)$ with $\vartheta(\lambda)$ as an equation connecting $\rho_p(\lambda)$ with $\vartheta(\lambda)$ and then taking a derivative of the resulting equation with respect to the chemical potential. Equation (82) takes the standard form (49) for a function $\vartheta^{-1}(\lambda)\partial \rho_p(\lambda)/\partial \mu$ and therefore the solution can be expressed with the help of the resolvent. We find

$$\frac{\partial \rho_p(\lambda)}{\partial \mu} = 2\pi \left( \rho_p(\lambda)\rho_h(\lambda) + \int_{-\infty}^{+\infty} d\alpha L(\alpha, \lambda)\rho_p(\alpha)\rho_h(\alpha) \right). \tag{83}$$

Therefore we obtain that

$$\frac{dn}{d\mu} = \int_{-\infty}^{+\infty} d\lambda \frac{\partial \rho_p(\lambda)}{\partial \mu} = 2\pi \int_{-\infty}^{+\infty} d\lambda \rho_p(\lambda)\rho_h(\lambda)\rho_t(\lambda) \tag{84}$$

where we used the relation (51) between the resolvent and the total density. This shows that

$$S(0) = \frac{dn}{d\mu}\bigg|_{\text{other Lagrange multipliers}} = \kappa_{\text{GGE}}. \tag{85}$$

Note that in the thermal case one usually rescales $\mu$ with the temperature as $\mu \to \mu/T$ which therefore leads to the known relation

$$S_T(0) = T\frac{dn}{d\mu}\bigg|_{\mu_2 = \frac{1}{T}} = T\kappa_{\text{T}}. \tag{86}$$

Hence we have proven equations (8) and (12).

In the case of the BEC interaction quench the relation between the density and the chemical potential on the GGE state is known

$$n = c\, e^{\mu/2}, \tag{87}$$

which immedietely gives that

$$S(0) = \frac{n}{2}. \tag{88}$$

Few comments about this result are in place. First of all $S(0)$ takes a finite value, this is because the GGE state after the quench, just like a thermal state, has a finite extensive entropy. This guarantees that the integral in (79) is non-zero. Second, the value $n/2$, as can be seen

from fig. 1, is rather typical. What really stands out is the independence of $S(0)$ of the post-quench interaction strength $c$. This independence is a direct consequence of the relation (87). The crucial feature of this relation is that the density is a linear function of the fugacity $z = \exp(\mu/2)$. Such dependence is typical of any equilibrium state where the chemical potential factorizes

$$\vartheta(\lambda) = \bar{\vartheta}(\lambda)e^{\mu/2}, \tag{89}$$

with $\bar{\vartheta}(\lambda)$ carrying dependence on all the chemical potentials associated with other conserved charges (if present) but independent of $\mu$. This is what happens for example in the classical limit of quantum free gases, where the Bose or Fermi statistics is replaced by the Boltzmann distribution

$$\frac{1}{e^{(\lambda^2-\mu)/T} \pm 1} \rightarrow e^{-(\lambda^2-\mu)/T}, \tag{90}$$

Therefore the relation $n \sim \exp(\mu/2)$ on the post-quench GGE state implies that after the quench the gas approaches an equilibrium state which exhibits a classical behavior when probed with density perturbations at small momenta.

## 5 Generalized detailed balance

Here we show how a generalization of the known detailed balance relation of the dynamic structure factor for thermal equilibrium allows to obtain the result (85) also for a non thermal state.

Let us consider system in a state described by a normalized GGE density matrix $\rho_{\lambda,\lambda} = \mathscr{Z}^{-1}e^{-\mathscr{F}_\lambda}$ with

$$\mathscr{F}_\lambda = \sum_j \mu_j \epsilon_\lambda^{(j)} - \mu N \tag{91}$$

and $\mathscr{Z} = \sum_\lambda e^{-\mathscr{F}_\lambda}$. The definition of the dynamic structure factor for a finite size system is

$$S(k,\omega) = \mathscr{Z}^{-1}(2\pi)^2 \sum_{\lambda,\lambda'} e^{-\mathscr{F}_\lambda} |\langle\lambda|\hat{\rho}(0)|\lambda'\rangle|^2 \delta(\omega-(E[\lambda]-E[\lambda']))\delta_{k,P[\lambda]-P[\lambda']}. \tag{92}$$

The reverse sign energy gives

$$S(k,-\omega) = \mathscr{Z}^{-1}(2\pi)^2 \sum_{\lambda,\lambda'} e^{-\mathscr{F}_\lambda} |\langle\lambda|\hat{\rho}(0)|\lambda'\rangle|^2 \delta(-\omega-(E[\lambda]-E[\lambda']))\delta_{k,P[\lambda]-P[\lambda']} \tag{93}$$

which by exchanging the sum over the eigenstates $\lambda$ and $\lambda'$ it can be rewritten as

$$S(k,-\omega) = \mathscr{Z}^{-1}(2\pi)^2 \sum_{\lambda,\lambda'} e^{-\mathscr{F}_\lambda} e^{-(\mathscr{F}_{\lambda'}-\mathscr{F}_\lambda)} |\langle\lambda|\hat{\rho}(0)|\lambda'\rangle|^2 \delta(\omega-(E[\lambda]-E[\lambda']))\delta_{k,P[\lambda]-P[\lambda']} . \tag{94}$$

We are now in position to take the thermodynamic limit. The sum over the eigenstates $|\lambda\rangle$ can be replaced into a functional integral over all the possible smooth $\vartheta(\lambda)$ and the saddle point of the Generalized Free energy selects only one contribution, given by the GTBA integral equation (22). We then recover the expression (38)[7]

$$S(k,\omega) = (2\pi)^2 \sum_{m=1}^\infty \sum_{(\mathbf{p},\mathbf{h})\in\mathscr{H}_\vartheta^{(m)}} |\langle\vartheta|\hat{\rho}(0)|\vartheta,\mathbf{h}\rightarrow\mathbf{p}\rangle|^2 \delta(q-k(\vartheta,\mathbf{p},\mathbf{h}))\delta(\omega-\omega(\vartheta,\mathbf{p},\mathbf{h})). \tag{95}$$

---

[7]For a more complete derivation of the thermodynamic limit of (92) see [29, 40]

Repeating the same procedure for the opposite energy case we find

$$S(k, -\omega) = (2\pi)^2 \sum_{m=1}^{\infty} \sum_{(\mathbf{p},\mathbf{h}) \in \mathscr{H}_{\vartheta}^{(m)}} e^{-\mathscr{F}(\mathbf{p},\mathbf{h})} |\langle \vartheta | \hat{\rho}(0) | \vartheta, \mathbf{h} \to \mathbf{p} \rangle|^2 \delta(q - k(\vartheta, \mathbf{p}, \mathbf{h})) \delta(\omega - \omega(\vartheta, \mathbf{p}, \mathbf{h})),$$

$$(96)$$

where $\mathscr{F}(\mathbf{p}, \mathbf{h})$ is the thermodynamic limit of the difference $\mathscr{F}_{\lambda'} - \mathscr{F}_{\lambda}$ given that the two states $\lambda'$ and $\lambda$ have the same density $\rho_p(\lambda)$ of rapidities and differ for only $m$ particle-hole excitations. In terms of thermodynamic Bethe Ansatz this is given by the dressed eigenvalues of the charges

$$\mathscr{F}(\mathbf{p}, \mathbf{h}) = \sum_j \mu_j \left( \epsilon_d^{(j)}(\mathbf{p}) - \epsilon_d^{(j)}(\mathbf{h}) \right) \tag{97}$$

where

$$\epsilon_d^{(j)}(\mathbf{p}) = \sum_{k=1}^{m} \left( \epsilon^{(j)}(p_k) - \int_{-\infty}^{+\infty} d\lambda \, \partial_\lambda \epsilon^{(j)}(\lambda) F(\lambda|p_k) \vartheta(\lambda) \right) \tag{98}$$

and

$$\epsilon_d^{(j)}(\mathbf{h}) = \sum_{k=1}^{m} \left( \epsilon^{(j)}(h_k) - \int_{-\infty}^{+\infty} d\lambda \, \partial_\lambda \epsilon^{(j)}(\lambda) F(\lambda|h_k) \vartheta(\lambda) \right). \tag{99}$$

In general the function $\mathscr{F}(\mathbf{p}, \mathbf{h})$ cannot be written in terms of $\omega(\vartheta, \mathbf{p}, \mathbf{h})$ and therefore cannot be taken out of the summations in (96). Therefore $S(k, \omega)$ and $S(k, -\omega)$ are not trivially related to each other. On the other hand in the limit $k \to 0$ we can use the fact that the sum over the number of excitations $m$ it saturated by the one particle-hole contribution $m = 1$. Therefore we can write for this case

$$\mathscr{F}(p, h) = \mathscr{F}(p) - \mathscr{F}(h) = \hat{\mathscr{F}}(k, \omega) + O(k^2) \tag{100}$$

where the relation $h(k, \omega)$ is given in (58). The function $\mathscr{F}(p, h)$ is now a function of $k$ and $\omega$ and it can be brought out of the summation over the excitations in (96) which then leads to

$$S(k \sim 0, -\omega) = e^{-\hat{\mathscr{F}}(k, \omega)} S(k \sim 0, \omega) + O\left(k^2\right) \tag{101}$$

where, given the mapping $h = h(k, \omega)$ in (58) we find

$$\hat{\mathscr{F}}(k, \omega) = \omega \left( \frac{\mathscr{F}'(h)}{2\pi v(h) \rho_t(h)} \right) \bigg|_{h=h(k,\omega)}. \tag{102}$$

We call (101) a generalized detailed balance. Note that when the density matrix of the system is the usual canonical ensemble the function $\mathscr{F}(\mathbf{p}, \mathbf{h})$ is a direct function of $\omega$

$$\mathscr{F}(\mathbf{p}, \mathbf{h}) = T^{-1} \left( \epsilon_d^{(2)}(\mathbf{p}) - \epsilon_d^{(2)}(\mathbf{h}) \right) = \omega/T \equiv \hat{\mathscr{F}}_T(k, \omega) \tag{103}$$

and the relation (101) is valid for any $k$ yielding the standard detailed balance [21]

$$S_T(k, -\omega) = e^{-\omega/T} S_T(k, \omega). \tag{104}$$

Relation (101), together with (85) leads to a generalized version of the compressibility sum rule, which in the thermal case reads [21]

$$\lim_{k \to 0} \int_{-\infty}^{\infty} \frac{d\omega}{2\pi} \frac{S_T(k, \omega)}{\omega} = \frac{\kappa_T}{2}. \tag{105}$$

To this end we use the generalized detailed balance to derive an alternative representation of the static structure factor. We can rewrite the detailed balance in the following way

$$S(k \sim 0, \omega) = \frac{\exp(\hat{\mathscr{F}}(k, \omega)/2)}{\exp(\hat{\mathscr{F}}(k, \omega)/2) - \exp(-\hat{\mathscr{F}}(k, \omega)/2)} (S(k \sim 0, \omega) - S(k \sim 0, -\omega)). \quad (106)$$

For the static structure factor we find

$$\begin{aligned}
S(k \sim 0) &= \int_{-\infty}^{\infty} \frac{d\omega}{2\pi} S(k \sim 0, \omega) = \frac{1}{2} \int_{-\infty}^{\infty} \frac{d\omega}{2\pi} (S(k \sim 0, \omega) + S(k \sim 0, -\omega)) \\
&= \frac{1}{2} \int_{-\infty}^{\infty} \frac{d\omega}{2\pi} \coth \frac{\hat{\mathscr{F}}(k, \omega)}{2} (S(k \sim 0, \omega) - S(k \sim 0, -\omega)) \\
&= \int_{-\infty}^{\infty} \frac{d\omega}{2\pi} \coth \frac{\hat{\mathscr{F}}(k, \omega)}{2} S(k \sim 0, \omega). \quad (107)
\end{aligned}$$

This relation is useful because it allows to connect the static structure factor with sum rules of the dynamic structure factor. In general at small momentum function $\hat{\mathscr{F}}(k, \omega)$ is also small which allows to expand the coth up to its leading order

$$S(k \sim 0) = 2 \int_{-\infty}^{\infty} \frac{d\omega}{2\pi} \frac{S(k \sim 0, \omega)}{\hat{\mathscr{F}}(k, \omega)}. \quad (108)$$

In the case of the thermal equilibrium we have

$$\hat{\mathscr{F}}_T(k, \omega) = \omega/T, \quad (109)$$

which, using the isothermal compressibility sum rule (105), leads to

$$S(0) = 2T \int_{-\infty}^{\infty} \frac{d\omega}{2\pi} \frac{S_T(k, \omega)}{\omega} = T\kappa_T. \quad (110)$$

On the other hand for the generic GGE case we find

$$\lim_{k \to 0} \int_{-\infty}^{\infty} \frac{d\omega}{2\pi} \frac{S_{\text{GGE}}(k, \omega)}{\hat{\mathscr{F}}(k, \omega)} = \frac{\kappa_{\text{GGE}}}{2} \quad (111)$$

which represents the generalized isothermal compressibility sum rule for a GGE equilibrium state.

As a non-trivial test of the validity of equation (12) we apply it to the case of the GGE state given by the BEC quench [34,35,42]. Then function $\hat{\mathscr{F}}(k, \omega)$ is known implicitly as a function of $p$ and $h$ and it reads

$$\begin{aligned}
\mathscr{F}(p, h) = \log\left(\frac{p^2}{c^2}\left(\frac{p^2}{c^2} + \frac{1}{4}\right)\right) - \log\left(\frac{h^2}{c^2}\left(\frac{h^2}{c^2} + \frac{1}{4}\right)\right) \\
- \int_{-\infty}^{+\infty} d\lambda \left(\frac{2}{\lambda} + \frac{8\lambda}{c^2 + 4\lambda^2}\right) \vartheta(\lambda) F(\lambda|p, h). \quad (112)
\end{aligned}$$

In the small momentum limit $p \sim h$ we obtain

$$\hat{\mathscr{F}}(k, \omega) = \frac{\omega}{2\pi v(h)\rho_t(h)} \left(\left(\frac{2}{h} + \frac{8h}{c^2 + 4h^2}\right) + \int_{-\infty}^{+\infty} d\lambda \left(\frac{2}{\lambda} + \frac{8\lambda}{c^2 + 4\lambda^2}\right) L(\lambda, h)\right) \quad (113)$$

where $h \equiv h(k, \omega)$ is given by solving (58). We consider the limit $c \to \infty$ limit where we can solve the mapping $h \equiv h(k, \omega)$ analytically. There we obtain

$$v(h) = 2h. \qquad \rho_t(h) = \frac{1}{2\pi} \qquad \text{where} \qquad h(k, \omega) = \frac{\omega}{2k} \tag{114}$$

which leads to $\hat{\mathscr{F}}(k, \omega) = 4k^2/\omega$. Therefore using the f-sum rule (36) we obtain

$$\lim_{c\to\infty} S_{\text{BEC}}(0) = 2 \int_{-\infty}^{\infty} \frac{d\omega}{2\pi} S(k, \omega) \frac{\omega}{4k^2} = \frac{n}{2}, \tag{115}$$

which indeed reproduces the result (85) and it can be proven that the same result holds for any value of the interaction $c$, given the generic form of $\hat{\mathscr{F}}(k, \omega)$ in (113).

## 6 Conclusion

In this work we compute an exact expression for the static structure factor $S(k)$ in the limit $k \to 0$ using the thermodynamic limit of the density form factors obtained in [40]. The result is obtained by a partial sum of such form factors. In particular only one particle-hole excitations are included, as they are the only relevant ones in the small momentum limit. A direct application of the result is the static structure factor of a gas at thermal equilibrium or in a generalized Gibbs ensemble obtained as the post-quench equilibrium state. We also shown that for the GGE ensemble $\lim_{k\to 0} S(k)$ is equal to a generalized isothermal compressibility, analogously to the thermal case. This is due to a generalized detailed balance that applies for any thermal and non-thermal equilibrium state in the small momentum regime of the density-density correlations.[8]

As a consequence of our results, we confirm a numerical observation from [42] that after the BEC interaction quench the zero momentum value of the time evolved static structure factor $S_t(k)$ in the infinity time limit is given by

$$\lim_{k\to 0} \lim_{t\to\infty} S_t(k) = \lim_{k\to 0} S_{\text{BEC}}(k) = n/2 \tag{116}$$

regardless of the final value of the interaction $c$. It would be interesting to understand this peculiar feature of this GGE saddle-point state and whether it generalizes to other interaction quenches in the Lieb-Liniger model.

The form factor used in this work can be expanded to higher orders in $k$ to compute the $O(k^2)$ part of the density-density correlations. Computing $S(k, \omega)$ for higher values of $k$ by including excitations with more particle-holes remains a challenging task that is postponed to future works. Such pursuit is highly relevant due to importance of the dynamic structure factor for the Bragg scattering experiments with 1d cold atomic gases [5–7]. On the other hand the small $k$ behavior of $S(k, \omega)$ also deserves a more extensive analysis, as it has recently shown to have surprising connections with the Kardar-Parisi-Zhang equation [53].

Finally the same thermodynamic limit of the form factor can be taken for other relevant operators with a determinantal form factor, as the bosonic annihilation/creation operator [16], or generic powers of it [54].

---

[8]Note that besides its connections to the density fluctuations, the compressibility is a relevant experimental observable. For example it plays an important role in quantum metrology providing a direct access to sensitive measurements of the chemical potential of a quantum gas [52].

Our analysis is limited to the Lieb-Liniger model but similar logic applies to other integrable models. In particular the generalized detailed balance relation should hold also for the two-point longitudinal spin correlations of integrable spin chains when the small momentum limit is dominated by the one particle-hole contribution (as it is the case for the XXZ model at finite magnetic field or temperature).

Recently it was proposed to employ a generalized fluctuation-dissipation relation to express chemical potentials of the GGE through the correlation functions [55]. Presented here computations might help in extending the applicability of this proposal to fully interacting theories.

## Acknowledgements

Authors are grateful to Pasquale Calabrese for a help in improving the manuscript and would like to thank Andrea Gambassi and Laura Foini for stimulating discussions. We kindly acknowledge SISSA (by the ERC under Starting Grant 279391 EDEQS) and ICTP for the hospitality during the early stage of this work.

**Funding information** The authors acknowledge support from LabEX ENS-ICFP:ANR-10-LABX-0010/ANR-10-IDEX-0001-02 PSL* (JDN) and from the NCN under FUGA grant 2015/16/S/ST2/00448 (MP).

## A  Zero temperature limit and zero extensive entropy states

The expression (41) if computed on the ground state of the Lieb-Liniger Hamiltonian gives a vanishing result. This happens for any distribution of rapidites where density of rapidities $\rho_p(\lambda)$ and holes $\rho_h(\lambda)$ have separate supports. This corresponds to states with zero Yang-Yang entropy (29) where the static structure factor takes a linear form in $k$

$$S(k) = \alpha|k| + O(k^2). \tag{117}$$

For such states we can compute the coefficient $\alpha > 0$. We start again from the expression (4.3)

$$S(k \sim 0) = 2\pi \int dh\, \rho_p(h)\rho_h\left(h + \frac{k}{2\pi\rho_t(h)}\right) \frac{|\langle\vartheta|\hat{\rho}|\vartheta, h \to h + \frac{k}{2\pi\rho_t(h)}\rangle|^2}{2\pi\rho_t(h)}. \tag{118}$$

We now chose a state with non-smooth distributions. Let us consider the ground state $\vartheta_{T=0}(\lambda) = \mathbf{1}_{-q_F < \lambda < q_F}$ with $q_F$ the Fermi momentum. Then the integration over $h$ is non-vanishing only where $\rho_p(h)\rho_h(h + kQ^{-1}(h)) \neq 0$, which is for $h > 0$ for positive $k$ and for $h < 0$ when $k$ is negative. For positive $k$ we have

$$S_{T=0}(k \sim 0) = 2\pi \int_0^\infty dh\, \rho_p(h)\rho_h\left(h + \frac{k}{2\pi\rho_t(h)}\right) \frac{|\langle\vartheta|\hat{\rho}|\vartheta, h \to h + \frac{k}{2\pi\rho_t(h)}\rangle|^2}{2\pi\rho_t(h)}. \tag{119}$$

The form factor can be expanded in $k$

$$\frac{|\langle\vartheta|\hat{\rho}|\vartheta, h \to h + \frac{k}{2\pi\rho_t(h)}\rangle|^2}{2\pi\rho_t(h)} = f_0(h) + kf_1(h) + O(k^2) \tag{120}$$

where $f_0(h) = 2\pi\rho_t(h)$ and $f_1(h)$ a smooth odd function that can be derived. The same can be done for the product of the two densities

$$\rho_p(h)\rho_h\left(h + \frac{k}{2\pi\rho_t(h)}\right) = \rho_p(h)\rho_h(h) + \frac{k}{2\pi\rho_t(h)}\rho_p(h)\partial_h\rho_h(h). \tag{121}$$

Therefore in the first order in $k$

$$S_{T=0}(k) = S(0) + (2\pi)k\left(\int_0^\infty dh\,\rho_p(h)\partial_h\rho_h(h) + \int_0^\infty dh\,\rho_p(h)\rho_h(h)\frac{f_1(h)}{2\pi\rho_t(h)}\right) + O(k^2). \tag{122}$$

Note that

$$\int_0^\infty dh\,\rho_p(h)\rho_h(h)\frac{f_1(h)}{2\pi\rho_t(h)} = 0 \tag{123}$$

since the the integrand is a smooth function of $h$ and $\rho_p(h)\rho_h(h)$ is non zero only in a set of measure zero. On the other hand $\rho_h(h) = \rho_t(h)\theta(h - q_F)$ which leads to

$$\int_0^\infty dh\,\rho_p(h)\partial_h\rho_h(h) = \rho_p(q_F)\rho_t(q_F) \tag{124}$$

and finally to a finite coefficient for the linear part of $S_{T=0}(k)$

$$S_{T=0}(k) = \left(2\pi\rho_p(q_F)\rho_t(q_F)\right)|k| + O(k^2) \tag{125}$$

where we restored the dependence on the sign of $k$. Using the definition of the Fermi velocity [9]

$$v_F = \frac{1}{2\pi\rho(q_F)\rho_t(q_F)} \tag{126}$$

we obtain then a known relation [21]

$$S_{T=0}(k) = \frac{|k|}{v_F} + O(k^2). \tag{127}$$

Our approach applies also to more general non-smooth states, as for example two split Fermi seas, as the one considered in [41]. There we have a state with the filling function given by $\vartheta(\lambda) = \mathbf{1}_{-q_2 < \lambda < -q_1} + \mathbf{1}_{q_1 < \lambda < q_2}$ with $q_1 < q_2$. Proceeding analogously as in the ground state case we find

$$S(k) = |k|\left(\frac{1}{v_{F_1}} + \frac{1}{v_{F_2}}\right) + O(k^2) \tag{128}$$

where $v_{F_i} = \frac{1}{2\pi\rho(q_i)\rho_t(q_i)}$.

# B  Resolvent

In this appendix we recall application of the resolvent technique to linear integral equations. In the context of thermodynamics of the Lieb-Liniger model it was first used in [45].

The integral equations are of the form

$$f(\lambda) - \int_{-\infty}^\infty d\lambda'\frac{\vartheta(\lambda')}{2\pi}K(\lambda - \lambda')f(\lambda') = g(\lambda), \tag{129}$$

which can be written in an operatorial form as

$$\left[(1 - \hat{K})f\right](\lambda) = g(\lambda), \tag{130}$$

where $\hat{K}$ acts on $\mathbb{R}^2$ as

$$\hat{K}(\lambda, \lambda') = \frac{\vartheta(\lambda')}{2\pi} K(\lambda - \lambda'). \tag{131}$$

We define the resolent $\hat{L}$ through the following equalities

$$(1 + \hat{L})(1 - \hat{K}) = 1 = (1 - \hat{K})(1 + \hat{L}). \tag{132}$$

The left part of the identity is equivalent to the integral equation

$$L(\lambda, \lambda') - \vartheta(\lambda') \int_{-\infty}^{\infty} d\alpha \frac{1}{2\pi} L(\lambda, \alpha) K(\alpha - \lambda') = \frac{1}{2\pi} K(\lambda - \lambda') \vartheta(\lambda'). \tag{133}$$

The combination $L(\lambda, \lambda') \vartheta^{-1}(\lambda')$ is invariant under exchange of $\lambda$ with $\lambda'$ which leads to the identity

$$\vartheta(\lambda) L(\lambda, \lambda') = \vartheta(\lambda') L(\lambda', \lambda) \tag{134}$$

and to the following integral equation for the resolvent

$$L(\lambda', \lambda) - \int_{-\infty}^{\infty} d\alpha \frac{\vartheta(\alpha)}{2\pi} L(\alpha, \lambda) K(\alpha - \lambda) = \frac{\vartheta(\lambda')}{2\pi} K(\lambda' - \lambda), \tag{135}$$

which follows also from the right hand side of the defining identity (132).

Important functions, namely the total density $\rho_t(\lambda)$ and the back-flow function $F(\lambda|p, h)$ have a simple representation in terms of the resolvent $L(\lambda, \lambda')$ that we have already shown in eqs. (51) and (52). Here we derive these formulas. Recall that $\rho_t(\lambda)$ obeys the following integral equation

$$\rho_t(\lambda) - \int_{-\infty}^{\infty} d\alpha \frac{\vartheta(\alpha)}{2\pi} K(\lambda - \alpha) \rho_t(\alpha) = \frac{1}{2\pi}. \tag{136}$$

Acting on both sides with $1 + L$ we find

$$\rho_t(\lambda) = \frac{1}{2\pi} \left( 1 + \int_{-\infty}^{\infty} d\alpha \, L(\lambda, \alpha) \right). \tag{137}$$

The back-flow function obeys

$$F(\lambda|p, h) - \int_{-\infty}^{\infty} d\lambda' \frac{\vartheta(\lambda')}{2\pi} K(\lambda - \lambda') F(\lambda'|p, h) = \frac{\theta(\lambda - p) - \theta(\lambda - h)}{2\pi}. \tag{138}$$

The right hand side can be written as

$$\frac{\theta(\lambda - p) - \theta(\lambda - h)}{2\pi} = -\int_{h}^{p} d\alpha \frac{K(\lambda - \alpha)}{2\pi}. \tag{139}$$

We again act on both sides with $1 + L$. The left hand side simply gives $F(\lambda, p, h)$. The right hand side is

$$-\int_{h}^{p} d\alpha \underbrace{\int d\lambda \left( \delta(\lambda' - \lambda) + L(\lambda', \lambda) \right) \frac{K(\lambda - \alpha)}{2\pi} \vartheta(\alpha)}_{(1+\hat{L})\hat{K}=-\hat{L}} \vartheta^{-1}(\alpha) = \int_{h}^{p} d\alpha \, \vartheta^{-1}(\alpha) L(\lambda', \alpha). \tag{140}$$

Therefore

$$F(\lambda|p, h) = -\int_{h}^{p} d\alpha \, \vartheta^{-1}(\alpha) L(\lambda, \alpha). \tag{141}$$

If a particle and hole are close to each other we get

$$F(\lambda|p, h) = -(p - h) \vartheta^{-1}(h) L(\lambda, h) + \mathcal{O}((p - h)^2). \tag{142}$$

There is also an equivalent expression following from the symmetry (134) of the resolvent

$$F(\lambda|p, h) = -(p - h) \vartheta^{-1}(\lambda) L(h, \lambda) + \mathcal{O}((p - h)^2). \tag{143}$$

## C  Ratio of Fredholm determinants

In this appendix we compute the ratio of the Fredholm determinants reported in (4.2). The formula is

$$\frac{\mathrm{Det}\left(1 - \frac{\vartheta}{2\pi}\left(K - \frac{2}{c}\right)\right)}{\mathrm{Det}\left(1 - \frac{\vartheta}{2\pi}K\right)}. \tag{144}$$

The kernel $K$ is symmetric and therefore the factor $\vartheta/(2\pi)$ can be freely interpreted in two ways

$$\left[\frac{\vartheta}{2\pi}K\right](\lambda, \lambda') = \frac{\vartheta(\lambda)}{2\pi}K(\lambda, \lambda') = K(\lambda, \lambda')\frac{\vartheta(\lambda')}{2\pi}, \tag{145}$$

without changing the determinant. The former equality is directly connected with the factor appearing in the definition (132) of the resolvent and is more convenient. To suggest such interpretation we write the factor $\vartheta/(2\pi)$ after the kernel $K$.

Taking formally the inverse of the Fredholm determinant, using the resolvent $L$ and multiplying the two infinitely dimensional matrices we obtain

$$\frac{\mathrm{Det}\left(1 - \left(K - \frac{2}{c}\right)\frac{\vartheta}{2\pi}\right)}{\mathrm{Det}\left(1 - K\frac{\vartheta}{2\pi}\right)} = \mathrm{Det}(1 + L)\,\mathrm{Det}\left(1 - \left(K - \frac{2}{c}\right)\frac{\vartheta}{2\pi}\right) = \mathrm{Det}\left(1 + (1 + L)\frac{\vartheta}{\pi c}\right). \tag{146}$$

We denote the new kernel

$$\mathscr{L} = (1 + L)\frac{\vartheta}{\pi c} \tag{147}$$

and its action on $\mathbb{R}^2$ is

$$\mathscr{L}(\lambda, \lambda') = \int_{-\infty}^{\infty} \mathrm{d}\alpha\,(\delta(\lambda - \alpha) + L(\lambda, \alpha))\frac{\vartheta(\lambda')}{\pi c} = \frac{1}{\pi c}\left(1 + \int_{-\infty}^{\infty} \mathrm{d}\alpha\,L(\lambda, \alpha)\right)\vartheta(\lambda'). \tag{148}$$

The factor in the bracket is the total density of the rapidites $2\pi\rho_t(\lambda)$ (see eq. (51)) and therefore

$$\mathscr{L}(\lambda, \lambda') = \frac{2}{c}\rho_t(\lambda)\vartheta(\lambda'). \tag{149}$$

The kernel $\mathscr{L}(\lambda, \lambda')$ is seperable into a function of $\lambda$ only times a function of $\lambda'$ only. Because of this the Fredholm determinant reduces to a trace

$$\mathrm{Det}(1 + \mathscr{L}) = 1 + \mathrm{Tr}\,\mathscr{L} = 1 + \int_{-\infty}^{\infty} \mathrm{d}\lambda\,\mathscr{L}(\lambda, \lambda). \tag{150}$$

The final result is

$$\frac{\mathrm{Det}\left(1 - \frac{\vartheta}{2\pi}\left(K - \frac{2}{c}\right)\right)}{\mathrm{Det}\left(1 - \frac{\vartheta}{2\pi}K\right)} = \mathrm{Det}(1 + \mathscr{L}) = 1 + \frac{2n}{c}, \tag{151}$$

where $n$ is the 1D density of the gas (27).

## D  Solution to the integral equation for $W(h, \lambda)$

In this appendix we solve the integral equation (74) for $W(h, \lambda)$ in terms of the total density $\rho_t(\lambda)$ and the resolvent $L(\lambda, \lambda')$. Due to linearity of the integral equation we can split $W(h, \lambda)$ in 2 parts that can be analyzed separately

$$W(h, \lambda) = -\frac{2b(h)}{c} \left(W_1(h, \lambda) + W_2(h, \lambda)\right), \tag{152}$$

where

$$W_1(h, \lambda) - \int_{-\infty}^{\infty} d\alpha \frac{\vartheta(\alpha)}{2\pi} W_1(h, \alpha) \left(K(\alpha - \lambda) - \frac{2}{c}\right) = 1, \tag{153}$$

$$W_2(h, \lambda) - \int_{-\infty}^{\infty} d\alpha \frac{\vartheta(\alpha)}{2\pi} W_2(h, \alpha) \left(K(\alpha - \lambda) - \frac{2}{c}\right) = -\frac{c}{2} K(h - \lambda). \tag{154}$$

The first function $W_1(h, \lambda) = W_1(\lambda)$ is $h$ independent. Comparing an equation for $W_1(\lambda)$ with an equation for $\rho_t(\lambda)$

$$2\pi \rho_t(\lambda) - \int_{-\infty}^{\infty} d\alpha \, \vartheta(\alpha) \rho_t(\alpha) K(\lambda - \alpha) = 1, \tag{155}$$

we make an ansatz

$$W_1(\lambda) = d_1 \rho_t(\lambda). \tag{156}$$

Substituting the ansatz into the integral equation we obtain

$$\frac{d_1}{2\pi} \underbrace{\left[2\pi \rho_t(\lambda) - \int d\alpha \, \vartheta(\alpha) \rho_t(\alpha) K(\lambda - \alpha)\right]}_{=1} + \frac{d_1}{2\pi} \frac{2}{c} \underbrace{\int_{-\infty}^{\infty} d\alpha \, \vartheta(\alpha) \rho_t(\alpha)}_{=n} = 1, \tag{157}$$

which gives

$$d_1 = \frac{2\pi}{1 + \frac{2n}{c}}. \tag{158}$$

Therefore

$$W_1(\lambda) = \frac{2\pi}{1 + \frac{2n}{c}} \rho_t(\lambda). \tag{159}$$

The second function $W_2(h, \lambda)$ is related to the resolvent and to the total density. The solution can be presented in two steps. We first use the resolvent to simplify the driving term. Observe that $\vartheta^{-1}(h) L(h, \lambda)$ obeys the following integral equation (coming from (133))

$$\vartheta^{-1}(\lambda) L(h, \lambda) - \int_{-\infty}^{\infty} d\alpha \frac{\vartheta(\alpha)}{2\pi} \left(\vartheta^{-1}(\alpha) L(h, \alpha)\right) K(\alpha - \lambda') = \frac{1}{2\pi} K(h - \lambda). \tag{160}$$

We define implicitly a new function $\bar{W}_2(h, \lambda)$

$$W_2(h, \lambda) = \bar{W}_2(h, \lambda) + d_2 \vartheta^{-1}(\lambda) L(h, \lambda). \tag{161}$$

Substituting it into the equation (154) for $W_2$ and using the above equation we find

$$\bar{W}_2(h, \lambda) - \int_{-\infty}^{\infty} d\alpha \frac{\vartheta(\alpha)}{2\pi} \bar{W}_2(h, \alpha) \left(K(\alpha - \lambda) - \frac{2}{c}\right) = -\left(\frac{d_2}{2\pi} + \frac{c}{2}\right) K(h - \lambda)$$
$$- \frac{2d_2}{c} \int_{-\infty}^{\infty} d\alpha \frac{1}{2\pi} L(h, \alpha). \tag{162}$$

We can fix now $d_2 = -\pi c$ to make the right hand side $\lambda$ independent. Moreover the integral over the resolvent is related to the total density $\rho_t(h)$ (see (51)). We obtain

$$\bar{W}_2(h,\lambda) - \int_{-\infty}^{\infty} d\alpha \frac{\vartheta(\alpha)}{2\pi} \bar{W}_2(h,\alpha) \left( K(\alpha - \lambda) - \frac{2}{c} \right) = 2\pi \rho_t(h) - 1. \tag{163}$$

This equation can be solved in the same way as the equation for $W_1$, the only difference is that the right hand side is now a function of $h$. We make an ansatz

$$\bar{W}_2(h,\lambda) = d_2(h) \rho_t(\lambda), \tag{164}$$

to find

$$d_2(h) = \frac{2\pi}{1 + \frac{2n}{c}} (2\pi \rho_t(h) - 1). \tag{165}$$

Therefore for $W_2(h,\lambda)$ we get

$$W_2(h,\lambda) = -\pi c \vartheta^{-1}(\lambda) L(h,\lambda) + \frac{2\pi}{1 + \frac{2n}{c}} (2\pi \rho_t(h) - 1) \rho_t(\lambda). \tag{166}$$

Recall that

$$W(h,\lambda) = -\frac{2b(h)}{c} (W_1(h,\lambda) + W_2(h,\lambda)), \quad b(h) = -\frac{\vartheta(h)}{2\pi L(h,h)}. \tag{167}$$

Therefore

$$W(h,h) = -1 + 2\pi \rho_t(h) \rho_p(h) \frac{2}{c} \left( 1 + \frac{2n}{c} \right)^{-1} L^{-1}(h,h). \tag{168}$$

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
