# Peer review of "Exact correlations in the Lieb-Liniger model and detailed balance out-of-equilibrium"

_SciPost Physics, doi:SciPost Phys. 1, 015 (2016)_

## Round 2 · Referee Report · Anonymous · 2016-12-3

Strengths
1. Very few results on GGE correlators exist. The paper is charting a new territory, with no landmarks;
2. Techniques used are difficult and demanding;
3. Given all the advances in quantum nonequilibrium dynamics, The subject is currently empirically relevant.
Weaknesses
1. No interpretation of the principal result is attempted. A hint on why the static structure factor of a quenched BEC does not depend on the interaction strength after quench, or, at least a discussion on other examples of results of such simplicity;
Report
This is an important paper, and it deserves publishing. The subject is correlation functions in representative eigenstates of the GGE ensemble, the correlators that produce values of observables in an integrable system after a relaxation form a non-equilibrium initial state. Very few results exist, all of them covering the one-point correlators. The manuscript in question is, to my knowledge, the first attempt to approach the two-point correlation functions. The object computed is the the infrared limit of the static structure factor, expressed through the rapidity distribution.
In an important particular case, a quench from a BEC, the commutation is performed all the way to an explicit result.
Results are important. In particular, they allow to predict the speed of sound in highly nonequilibrium integrable gases, something that can be measured. Furthermore, the techniques developed do apply to spin systems. In general, the paper is a valuable contribution to the field of quantum nonequilibrium dynamics.
Requested changes
Besides a few trivial suggestions listed below, I have one that is difficult to implement.
1.1. The $n/2$ authors obtain for the zero-momentum static structure factor of a BEC, independently of the interaction strength, suggests a deed fundamental interpretation. A minimal discussion is in order. The very least, the sign of the effect must be discussed: one would naively expect that thanks to the interatomic repulsion, the structure factor would go down from the its zero value at the non-interacting BEC. Yet, it goes up.
Minor suggestions:
2.1. Using the same letter for both $S(x,\, t)$ and $S(k)$ may be misleading;
2.2. $\Big|_{\{\mu_i\}_i}$ is extremely misleading. Should be something like $\Big|_{\mbox{other Lagrange multipliers}}$;
2.3. Presence of $T$ in the definition (7), but absence of any Lagrange multipliers in (9) looks like an inconsistency;
2.4. Around Eq. 11, some discussion of the relationship between the notion of the "detailed balance" used and the common
notion notion of the "detailed balance", that implies absence of loops in the probability/matter transfer in a steady state, is needed;
2.5. Fig. 1 is never referenced;
2.6. The order in which energy and momentum appear In Eq.15 is different from the one in the preceding sentence;
2.7. In Eq. 25, energy is defined as a momentum and momentum as energy;
2.8. Around Eq. 27, it should be mentioned, that $m$ stands for the number of the particle-hole excitations, it has never been defined;
2.9. Some interpretation of the function $F(\lambda|p,\,h)$ is needed. Is it the deformation of the rapidity distribution due to a particle-hole excitation?

---

## Round 3 · List of Changes

We thank the referee for carefully reading our manuscript and kindly proposing important improvements to it.

1.1 We added a short discussion on this result at the end of section 4.3 and we added fig. 4 highlighting the dependence of S(0) on the rapidity distribution. The figure shows that at the intuitive level what controls S(0) is not the interatomic repulsion per se, but the amount of the entropy in the rapidity distribution. In practice what matters is the compressibility and for the BEC quench it takes a very simple form reported in eqs. (87) and (88). The remarkable phenomena is that in this limit S(0) is independent of the post-quench value of the interaction. At the level of equations the reason for this is clear. At the level of intuitions we have only a limited explanation presented at the end of section 4.3.

2.1 We added a bar for S(x,t) in order to not confuse it with its Fourier transform S(k,omega).
2.2 We implemented the suggested change.
2.3 We would like to point out that footnote 3 explains this only apparent discrepancy in detail.
2.4 We added equation (9) that explains what is detailed balance for a quantum gas in a thermal state.
2.5 We now refer to it.
2.7 We thank the referee for spotting this mistake, it's now corrected.
2.8 We now define m as the number of particle-hole.
2.9 We remark that this is indeed the shift due to particle-hole excitations on the rapidity distribution.

You are currently on this page

Resubmission 1611.00194v3 on 20 December 2016

---

## Editorial Decision

published